# A mathematical model for ketosis-prone diabetes suggests the existence of multiple pancreatic β-cell inactivation mechanisms

Sean A Ridout[1,2]*, Priyathama Vellanki[3,4], Ilya Nemenman[1,2,5]*

[1]Department of Physics, Emory University, Atlanta, United States; [2]Initiative in Theory and Modeling of Living Systems, Emory University, Atlanta, United States; [3]Department of Internal Medicine, Division of Endocrinology, Emory University School of Medicine, Emory University, Atlanta, United States; [4]Grady Health System, Atlanta, United States; [5]Department of Biology, Emory University, Atlanta, United States

## eLife Assessment

This theoretical study makes a **useful** contribution to our understanding of a subtype of type 2 diabetes – ketosis-prone diabetes mellitus (KPD) – with a potential impact on our broader understanding of diabetes and glucose regulation. The article presents an ordinary differential equation-based model for KPD that incorporates a number of distinct timescales – fast, slow, as well as intermediate, incorporating a key hypothesis of reversible beta cell deactivation. The presented evidence is **solid** and shows that observed clinical disease trajectories may be explained by a simple mathematical model in a particular parameter regime.

**\*For correspondence:**
sridout@emory.edu (SAR);
ilya.nemenman@emory.edu (IN)

**Abstract** Ketosis-prone diabetes mellitus (KPD) is a subtype of type 2 diabetes, which presents much like type 1 diabetes, with dramatic hyperglycemia and ketoacidosis. Although KPD patients are initially insulin-dependent, after a few months of insulin treatment, roughly 70% undergo near-normoglycemia remission and can maintain blood glucose without insulin, as in early type 2 diabetes or prediabetes. Here, we propose that these phenomena can be explained by the existence of a fast, reversible glucotoxicity process, which may exist in all people but be more pronounced in those susceptible to KPD. We develop a simple mathematical model of the pathogenesis of KPD, which incorporates this assumption, and show that it reproduces the phenomenology of KPD, including variations in the ability for patients to achieve and sustain remission. These results suggest that a variation of our model may be able to quantitatively describe variations in the course of remission among individuals with KPD.

## Introduction

Diabetes, a disease characterized by high blood glucose levels, is one of the most common chronic diseases in the United States, affecting more than 34 million adults in 2020 (*CDC, 2020*). In healthy people, blood glucose levels are regulated by the hormone insulin through a negative feedback loop. High blood glucose promotes the secretion of insulin from the β-cells of the pancreas. Insulin, in turn, lowers blood glucose by suppressing glucose production (e.g., in the liver) and promoting glucose

uptake (e.g., by the muscle). Thus, diabetes is associated with defects in this homeostatic feedback loop (**DeFronzo, 1988**). Diabetes is often divided into type 1 diabetes (T1D) and type 2 diabetes (T2D). In T1D, autoimmune destruction of the β-cells results in a complete lack of insulin production, and thus dependence on insulin treatment for survival (**Katsarou et al., 2017**). In T2D, such auto-immunity is not present. T2D is generally viewed as a progressive disease, in which blood glucose control gradually worsens over the course of years. In the early stages of T2D pathogenesis, the body produces insulin, but its ability to reduce blood glucose levels is inadequate (*insulin resistance*). In later stages of T2D, insulin production declines, eventually leading to dependence on insulin treatment (**DeFronzo et al., 2015**).

Mathematical modeling of diabetes has focused on two widely separated timescales. Short-timescale models describe the dynamics of insulin and glucose over minutes or hours (**Bergman et al., 1979**; **Cobelli et al., 2014**; **Toffolo et al., 1995**; **Toffolo et al., 2001**; **Stefanovski et al., 2020**). These models describe a negative feedback loop, where glucose promotes insulin secretion and insulin promotes glucose disposal. Such models have been used extensively in the interpretation of physiological measurements, for example, to infer insulin sensitivity on the basis of glucose tolerance tests (**Bergman et al., 1979**; **Dalla Man et al., 2002**). A second class of models integrates simple short-timescale models with processes that occur on longer timescales of months to years, with the goal of describing T2D pathogenesis (**Topp et al., 2000**; **Ha et al., 2016**; **Ha and Sherman, 2020**; **Karin et al., 2016**; **De Gaetano and Hardy, 2019**). These models generally include an *adaptive* process, where either slightly elevated glucose or per-β-cell insulin secretion prompts increases in β-cell activity or mass, and a *glucotoxicity* process, where severely elevated glucose causes death or deactivation of β-cells. Adaptation may involve increases in mass or secretory function of individual cells or it may include cell division that can eventually produce recovery from death of cells (**Topp et al., 2000**; **Ha et al., 2016**), but evidence suggests that -cells do not proliferate appreciably in adults (**Cnop et al., 2010**; **Saisho et al., 2013**). Whether or not some amount of proliferation can occur, it is empirically true that T2D of short duration can generally be reversed by interventions which improve insulin resistance and secretion, while T2D of longer duration which has advanced to severe hyperglycemia cannot be reversed (**UK Prospective Diabetes Study Group, 1995**; **Turner, 1999**), a fact which is explained by existing models since they connect severe hyperglycemia to severe losses of β-cells (**Ha et al., 2016**).

The focus of this work is on *ketosis-prone type 2 diabetes* (KPD), a subtype of T2D, common in particular in patients of African descent (**Vellanki and Umpierrez, 2017**; **Umpierrez et al., 1995**; **McFarlane et al., 2001**; **Mauvais-Jarvis et al., 2004b**). KPD presents with ketosis or diabetic keto-acidosis (DKA), an episode of extremely high blood glucose and low blood pH due to excess ketone

**Table 1.** Features of KPD that a mathematical model should explain.

| 1 | Associated with obesity and insulin resistance (**Umpierrez et al., 1995**) |
|---|---|
| 2 | Potentially rapid onset of hyperglycemic crisis (**Umpierrez et al., 1995**; **Vellanki and Umpierrez, 2017**) |
| 3 | Onset of hyperglycemic crisis associated with period of high sugar consumption (this is a common observation in our (PV) clinic, but it is difficult to test its statistical significance because antecedent data for new-onset diabetes are usually unavailable) |
| 4 | Ketotic relapses are preceded by a period of hyperglycemia (**Mauvais-Jarvis et al., 2004b**) |
| 5 | Poor β-cell function at presentation (**Umpierrez et al., 1995**) |
| 6 | Improvement of β-cell function with weeks or months of insulin treatment (**Umpierrez et al., 1995**; **McFarlane et al., 2001**) |
| 7 | Worsening of β-cell function after many hours of glucose infusion (**Umpierrez et al., 2007**) |
| 8 | Patient-to-patient variation in ability to achieve and sustain remission (**Mauvais-Jarvis et al., 2004b**; **McFarlane et al., 2001**; **Vellanki et al., 2016**; **Umpierrez et al., 1997**; **Vellanki et al., 2020**) |
| 9 | Duration of remission is improved by (non-insulin) blood glucose management due to greater β-cell function (**Vellanki et al., 2016**; **Umpierrez et al., 1997**; **Vellanki et al., 2020**) |
| 10 | Long-term declines in insulin sensitivity and secretion are similar to 'conventional' type 2 diabetes (**Mauvais-Jarvis et al., 2004b**; **Banerji et al., 1996**) |
| 11 | Patients with less insulin resistance have lower glucose in remission, although it is unclear if this predicts long-term prognosis (**Vellanki et al., 2020**) |

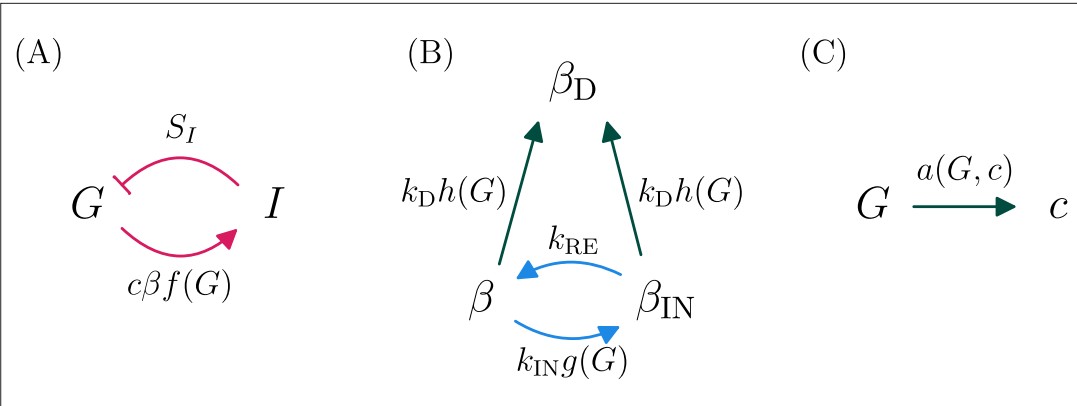

**Figure 1.** A model of ketosis-prone diabetes mellitus incorporating the hypothesis of reversible β-cell deactivation. Here we illustrate schematically the model described by *Equations 1–5* ('Materials and methods'). (**A**) Glucose promotes insulin secretion, and insulin lowers glucose, in a negative feedback loop. The effect of insulin on glucose disposal is mediated by the insulin sensitivity $S_I$, and the amount of insulin secreted is proportional to the mass of active beta cells $\beta$ and the per-cell secretion rate $c$. This couples these fast dynamics to the slower dynamics in panels (**B, C**). (**B**) Longer timescale dynamics incorporate two distinct types of glucotoxicity. An intermediate scale (days–weeks), reversible glucotoxicity with rate $k_{IN}g(G)$ produces an inactive pheonotype $\beta_{IN}$, which recovers at rate $k_{RE}$. Simultaneously, a slower process causes permanent β-cell deactivation or death at rate $k_D h(G)$. The dependence of these rates on $G$ couples these processes to those in panel (**A**). (**C**) The per-β-cell secretion rate $c$ adapts to maintain euglycemia, unless $c$ reaches its maximum value $c_{max}$. Pink arrows indicate fast (minutes–hours) processes, blue arrows intermediate-rate processes (days–weeks), and green slow processes (months–years).

bodies. Such an episode is associated with a lack of insulin secretion and action and thus, in the absence of a known precipitating cause, is classically thought of as a symptom of T1D rather than T2D. Unlike in T1D, however, β-cell autoimmunity is not present in KPD. Further, KPD patients can often achieve partial remission: after a few weeks or months of treatment with insulin, they are no longer dependent on insulin and can control blood glucose with diet or oral antidiabetic agents (*Mauvais-Jarvis et al., 2004b*; *McFarlane et al., 2001*; *Vellanki et al., 2016*; *Umpierrez et al., 1997*; *Vellanki et al., 2020*). There is substantial heterogeneity in response to insulin treatment. Firstly, some patients are unable to achieve remission. Secondly, when remission is achieved, it does not always last: the duration of the remission (the time until a patient's condition worsens to the point where they again require insulin) varies from 6 months to 10 years (*Mauvais-Jarvis et al., 2004b*). Although no study has systematically characterized the duration of high blood glucose preceding the acute DKA event, patients generally report less than 4 weeks of diabetes-associated symptoms (polyuria, polydipsia, and weight loss) prior to the emergency (*Vellanki and Umpierrez, 2017*). This does not, however, rule out a longer period of slightly elevated blood glucose, as in pre-diabetes. In *Table 1*, we list these and other known features associated with KPD, which any mathematical model of KPD should be able to explain.

Existing models of diabetes pathogenesis, in which β-cell deactivation occurs slowly over months and is often thought to be irreversible, cannot account for these phenomena. Instead, the possibility of rapid onset of severe hyperglycemia and rapid remission of KPD suggests the existence of a faster pathogenic mechanism, operating on timescales of days or weeks. Indeed, in an experiment where a KPD patient was given a 20-hour infusion of glucose, it was observed that their level of insulin secretion (but not that of a control) dropped by about a factor of two over the course of the infusion (*Umpierrez et al., 2007*). (Note that quantitative interpretation of this experiment is hard since there was just one patient and since the experiment measured the ratio of C-peptide to glucose, and not directly the amount of secreted insulin at fixed glucose.) This intermediate-timescale process (occurring over a period of a few days) is not in current mathematical models of T2D. We thus propose that KPD differs from conventional T2D through the presence of a second type of glucotoxicity, which is fast but reversible. This mechanism may also be present, but merely less pronounced, in T2D patients who do not show features of KPD. Adding this mechanism to the existing view of diabetes

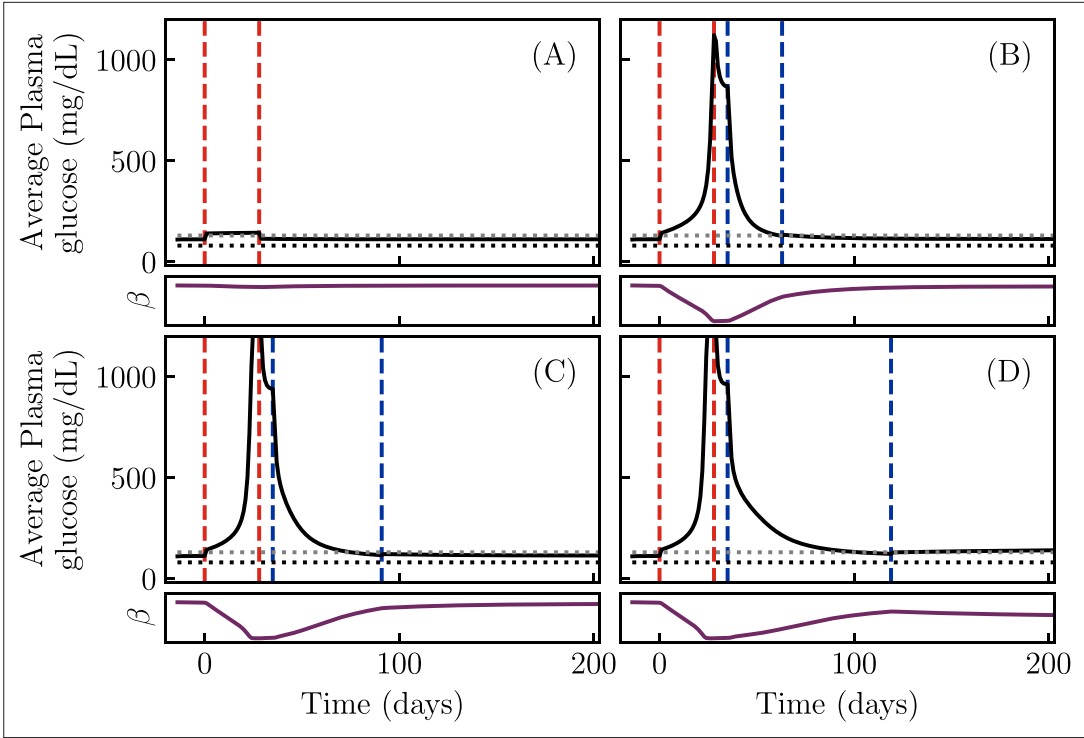

**Figure 2.** Simulations showing the onset and remission of ketosis-prone diabetes mellitus in our model. Solid black lines show daily blood glucose averages. Red dashed lines demarcate a period of increased sugar consumption, while blue dashed lines demarcate a period of insulin treatment, if present. Horizontal dotted lines indicate 80 mg/dL (black, typical normoglycemia) and 130 mg/dL (gray, diabetes control threshold per the American Diabetes Association). (**A**) With a low rate of reversible β-cell inactivation, blood glucose returns to normal after the period of high sugar consumption. ($k_{RE} \approx 0.04 \text{ day}^{-1}, k_{IN} \approx 4.8 \text{ day}^{-1}, k_D = 10^{-3} \text{ day}^{-1}$.) (**B**) With a high rate $k_{IN}$ of reversible β-cell inactivation, the same period of high sugar consumption produces a sharp rise in the blood glucose level, due to a sharp drop in β-cell function (purple curve), which persists after the period of sugar consumption ends. A sufficiently long period of insulin treatment can produce an insulin-free remission. ($k_{IN} \approx 71 \text{ day}^{-1}$, other parameters same as **A**). (**C**) A lower rate $k_{RE}$ of β-cell reactivation, compared to (**B**), increases the time required to produce remission with the same insulin treatment. ($k_{RE} \approx 0.028 \text{ day}^{-1}$, other parameters same as B) (**D**) A higher rate $k_D$ of permanent β-cell death, compared to (**C**), results in a failure to achieve insulin-free remission. ($k_D \approx 0.07 \text{ day}^{-1}$, other parameters same as **C**). Note that this is due to a permanent loss of β-cell function (purple curve). All model parameters other than those of glucotoxicity and total β-cell mass are fixed at values given in 'Materials and methods'. Also note that the rate of reversible glucotoxicity is much slower than the value of $k_{IN}$ might suggest because $g(G)$ is very small at physiological glucose values (e.g., $g(100 \text{ mg/dL}) \approx 0.0006$).

pathogenesis, we arrive at a mathematical model depicted in *Figure 1* and described by *Equations 1–5* ('Materials and methods').

Below we only simulate patients in advanced prediabetes, where we presume that β-cells have already maximized their secretory capacity, so that the adaptive process in *Figure 1C* is saturated. Further details are provided in 'Materials and methods'.

## Results

## Numerical simulations of this model reproduce the phenomenology of KPD

We begin by simulating our model, *Equations 1–5*, for different parameter choices, representing different possible patient phenotypes. Since the glycemic emergency or DKA in KPD is often preceded by a period of high sugar consumption (*Table 1, criterion 3*), and in past experiments such patients show reductions in β-cell function during a period of glucose infusion (*Umpierrez et al., 2007*), we

simulate a period of a month of increased sugar consumption. The outcome of this simulation is shown for four different choices of model parameters in *Figure 2*. All four sets of parameter values are chosen to result in a similar level of fasting hyperglycemia ($G \approx 110 \, \mathrm{mg/dL}$) by adjusting the values of $k_{\mathrm{IN}}$, $k_{\mathrm{RE}}$, $k_{\mathrm{D}}$, and $\beta_{\mathrm{TOT}}$ with all other parameters held fixed.

*Figure 2A* shows the daily average of the plasma glucose for a low value of the reversible β-cell inactivation rate $k_{\mathrm{IN}}$. For this set of parameters, although blood glucose rises during the period of high sugar consumption, it returns to its previous value when the period of high sugar consumption ends. In contrast, when $k_{\mathrm{IN}}$ is large (*Figure 2B*), the same period of increased sugar consumption results in a dramatic rise of blood glucose which persists after the sugar consumption is stopped (between the second red and first blue lines). Insulin treatment, carried out until blood glucose is well controlled (between the blue lines), produces a remission of this dramatic hyperglycemia, returning the glucose to its value before the period of high sugar consumption. The character of this remission is affected by the parameters of the model. *Figure 2C* shows a simulation with a slower reactivation rate $k_{\mathrm{RE}}$. In this case, a longer period of insulin treatment is necessary in order to produce remission. Finally, *Figure 2D* shows a simulation with this lower reactivation rate, and furthermore a higher rate of β-cell death $k_{\mathrm{D}}$. As a result, the total β-cell population, $\beta_{\mathrm{TOT}}$, declines enough during the period of high blood glucose that insulin-free remission cannot be maintained.

We thus see that our model reproduces the qualitative features associated with KPD. Starting from a period of mild hyperglycemia, a period of increased sugar consumption can produce a rapid progression to reduced β-cell function and severe hyperglycemia (*Table 1, criteria 2-5*). Treatment with insulin for weeks or months can produce a partial remission, in which a state of mild hyperglycemia is maintained without further insulin treatment (*Table 1, criterion 6*). Further, the model can produce heterogeneous patient outcomes: the time required to achieve remission and the possibility of relapse are different for different parameter values (*Table 1, criterion 8*). Since a failure to maintain remission emerges from glucotoxicity in this model, it is also consistent with the observation that better blood-glucose management increases the duration of remission (*Table 1, criterion 9*). Since long-term prognosis, in the absence of further crises, is governed by the same slow glucotoxicity process as progression of conventional T2D, criterion 10 is also satisfied. Criteria 1 and 11 will be discussed below. Finally, recall that criterion 7 has been directly included as an assumption in the model. Thus, the model reproduces the desired features of KPD, described in *Table 1*.

We will now perform a more detailed analysis of the properties of our model to understand why it produces this phenomenon of rapid onset and remission and which biological parameters are expected to be most relevant to the KPD phenotype.

## Rapid onset and remission of severe hyperglycemia result from bistability of fasting glucose

The onset and remission seen in our simulations are reminiscent of bistability in dynamical systems: the blood glucose in the KPD patients in this model seems to have two stable values, with transitions between the two triggered by intense glucose consumption or insulin treatment. Indeed, since bistability is often associated with positive feedback loops (*Alon, 2019*), and the reversible inactivation of β-cells by glucose creates such a loop, it is natural to expect such a behavior. We now make this intuition more precise.

To simplify analyses in this section, we will study constant exogenous glucose intake $M(t) = M$; results using separate meals (as in our simulations, *Figure 2*) are similar.

Since we assume that permanent β-death is slow (occurring over months or years), we may understand the behavior over shorter timescales (weeks) by neglecting it (for our form of death dynamics, the only true fixed point is at $\beta = 0$). We then solve for the fixed points (where $\beta$, $\beta_{\mathrm{IN}}$, $c$, $G$, and $I$ are not changing) of *Equations 1–5*.

The result for varying values of $M$ and $F_I$, and for the patient-specific parameters from *Figure 2A and B*, is shown in *Figure 3*. Note that we have included (impossible) negative values of $M$ and $F_I$ (shown with gray background) for conceptual clarity.

The curves for the non-KPD simulation (panels A1, A2) are *monostable*: regardless of sugar consumption or insulin infusion, they show only a single fixed point value of $\beta$ or $G$. Thus, the patient in *Figure 2A* has a single well-defined fasting glucose (at $M = 0$), which is independent of past glucose consumption.

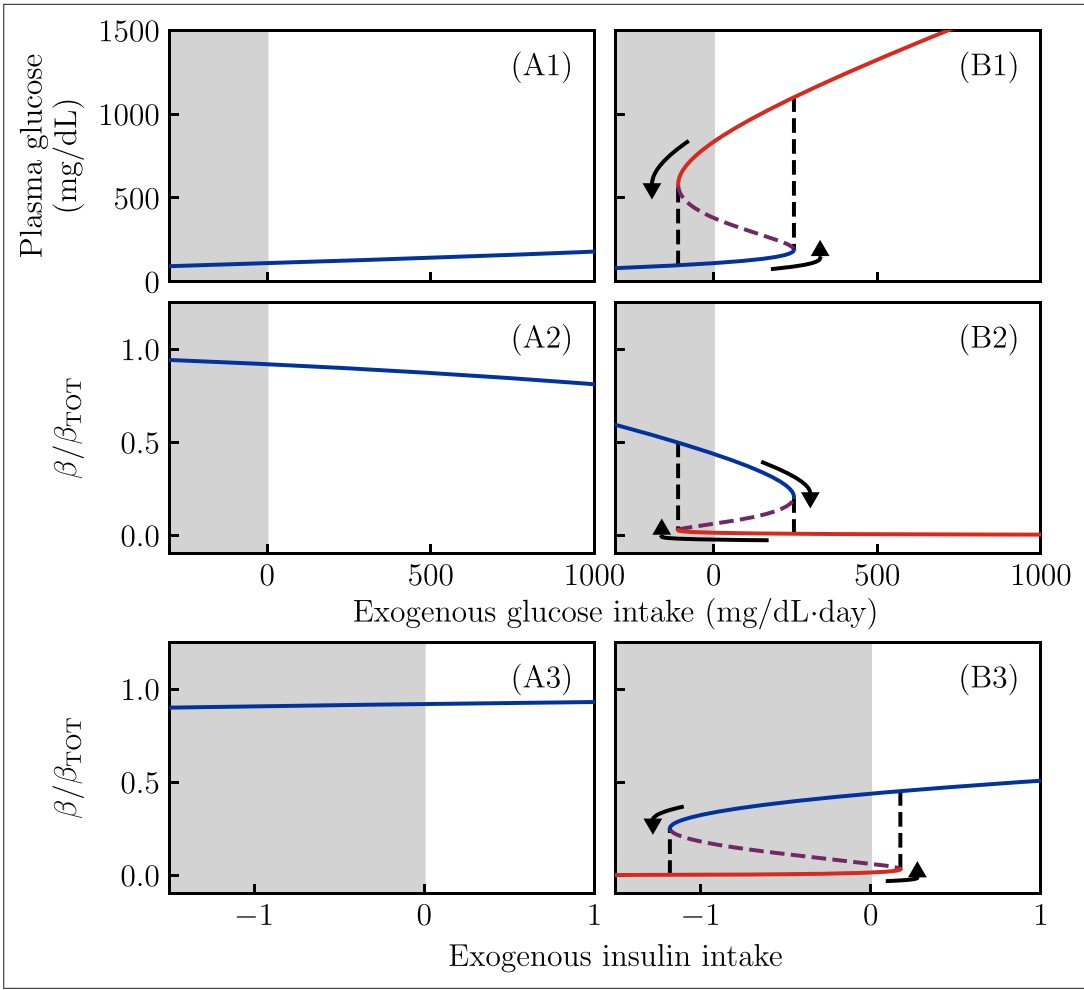

**Figure 3.** Fixed points of the dynamics as a function of sugar intake and exogenous insulin flux for all other parameter values as in *Figure 2A and B*. Gray regions denote negative exogenous fluxes, which are not possible in reality. (**A1, A2, A3**) Regardless of glucose or insulin intake, the non-ketosis-prone diabetes mellitus (KPD) parameter values give a single steady-state value for glucose (**A1**) and $\beta$ (**A2, A3**). (**B1, B2**) At KPD parameter values, at zero glucose intake, there are two possible stable values for glucose and $\beta$, a state with moderate $G$ and high $\beta$ (blue), and a state with high $G$ and low $\beta$ (red). Both of these states can stably persist without exogenous glucose or insulin intake. Either a sufficiently large glucose flux (**B1, B2**) or an (impossible) negative insulin flux (**B3**) can push the system past a bifurcation (black dashed line), forcing the system into the low-$\beta$ fixed point. Similarly, a negative glucose intake or sufficiently large insulin treatment pushes the system past a bifurcation, which forces it into the high-$\beta$ fixed point. Purple dashed lines show the unstable fixed point, which is present in the bistable region.

The curves for the KPD simulation (panels B1, B2), in contrast, show *bistability*. Even in the absence of glucose consumption ($M = 0$), there are two stable states: a high-$\beta$, low-$G$ state (blue) and a low-$\beta$, high-$G$ state (red). These two states are separated by an unstable fixed point (purple); values of $\beta$ above this point tend to flow toward the healthy fixed point, while values below it tend to flow toward the hyperglycemic fixed point.

When $M$ is increased, the system passes through a bifurcation; for large values of $M$ (right of the black dashed lines) only the hyperglycemic fixed point is present. Thus, when sugar consumption is too high for a long period of time, the patient will enter the hyperglycemic state (arrow) even if they began in the healthy state. Since both states are stable at $M = 0$, the hyperglycemic state will persist even if the high sugar consumption stops. Note, however, that for sufficiently *negative M* (which is not possible in reality), the hyperglycemic fixed point disappears, and thus the system would be driven back into the healthy fixed point.

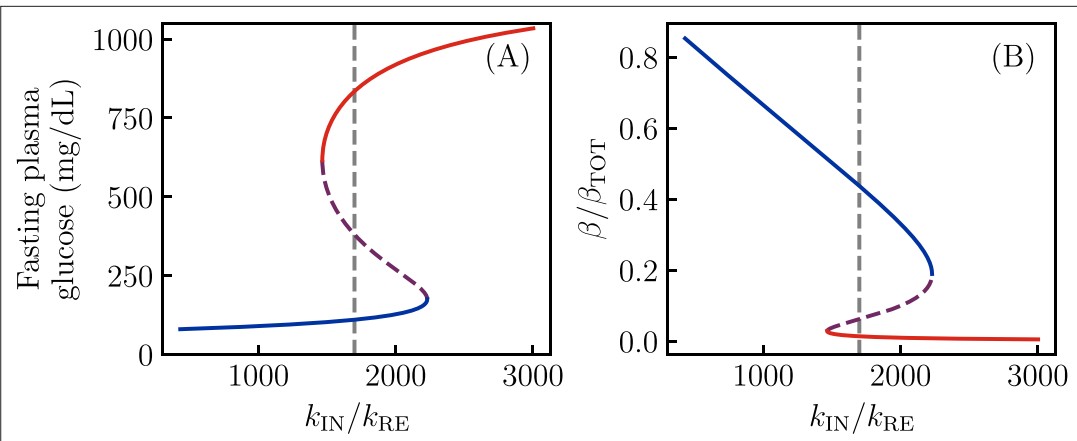

**Figure 4.** Fixed points as a function of reversible inactivation rate. All parameters of the model are taken to be as in *Figure 2B*, but now the ratio of reversible β-cell inactivation and reactivation rates is varied. As before, red lines denote the hyperglycemic state. (**A**) Fasting glucose as a function of the inactivation rate, showing that there is a range of values over which bistability, and thus ketosis-prone diabetes mellitus-like phenomena, is observed. (**B**) The same phenomena are seen in bistability of the active β-cell fraction.

This behavior is a classic 'toggle switch' (*Gardner et al., 2000*). In this model, for parameter values that predispose to KPD, both the high-$\beta$ and low-$\beta$ fixed points are stable, and moving from one to the other requires a sufficiently large 'push'.

The purple dashed lines show the unstable fixed point, which appears in the bistable region. Since the dynamics of glucose are much faster than those of $\beta$, the unstable fixed point value of $\beta$ acts roughly like a threshold: a patient with β above the purple line would, in the absence of treatment or changes of glucose intake, move toward the high-$\beta$ fixed point, while one with $\beta$ below the purple line will continue to decline until reaching the low-$\beta$ fixed point.

Since negative glucose consumption is not possible, in reality the 'push' out of the low-$\beta$ fixed point must come from insulin infusion. Panels (A3, B3) show the fixed-point values of $\beta$ as a function of insulin infusion rate $F_I$ (again, negative values are not possible in reality). Again, when parameters do not predispose to KPD (panels A3) there is a single fixed point for all $F_I$. For parameters predisposing to KPD (panel B3), there are two stable fixed points at $F_I = 0$, separated by an unstable fixed point. A sufficiently large insulin infusion rate $F_I$ pushes the system past a bifurcation (right black dashed line) such that, during the insulin treatment, there is only a single fixed point, which has high $\beta$. Since both states are stable at $F_I = 0$, if the insulin treatment lasts long enough for the healthy (blue) fixed point to be reached, it will persist even after insulin treatment stops.

The important parameter which distinguishes a predisposition to 'conventional' T2D (*Figure 2A*) from one to KPD (*Figure 2B*) is the ratio $k_{IN}/k_{RE}$ of the reversible β-cell inactivation and reactivation rates. In *Figure 4*, we show the effect of this parameter on the (fasting) fixed points of the model. For this particular degree of insulin sensitivity, total β-cell number, etc., we see there is a threshold value of $k_{IN}/k_{RE}$ below which no bistability, and thus no KPD-like phenomena, is seen. Similarly, there is a second threshold above which there is no healthy state, and thus a complete loss of β-cell function.

As in *Figure 2D*, it is possible for the irreversible β-cell death during a period of high glucose to move the fixed points, or perhaps even destroy the high-$\beta$ fixed point, resulting in permanent insulin dependency. Furthermore, improvements in insulin sensitivity $S_I$, associated with lifestyle modifications, may be able to destroy the low-$\beta$ fixed point, or increase the degree of sugar intake required to destabilize the high-$\beta$ fixed point. To study these possibilities, we adjust $S_I$ and the total β-cell population $\beta_{TOT} = \beta + \beta_{IN}$, with all other parameters of the model held fixed to the values from *Figure 2A and B*.

Firstly, we vary $S_I$ and $\beta_{TOT}$ individually. As shown in *Figure 5*, the KPD-like parameter values (panels B1, B2) again produce bifurcations as either $S_I$ or $\beta_{TOT}$ is varied. Thus, improvements in insulin sensitivity can produce a partial remission, and death of β-cells can prevent remission. Again, the set of parameters with a low reversible deactivation rate (panels A1, A2) show no bistability regardless of $S_I$ or $\beta_{TOT}$—they only permit 'conventional' T2D.

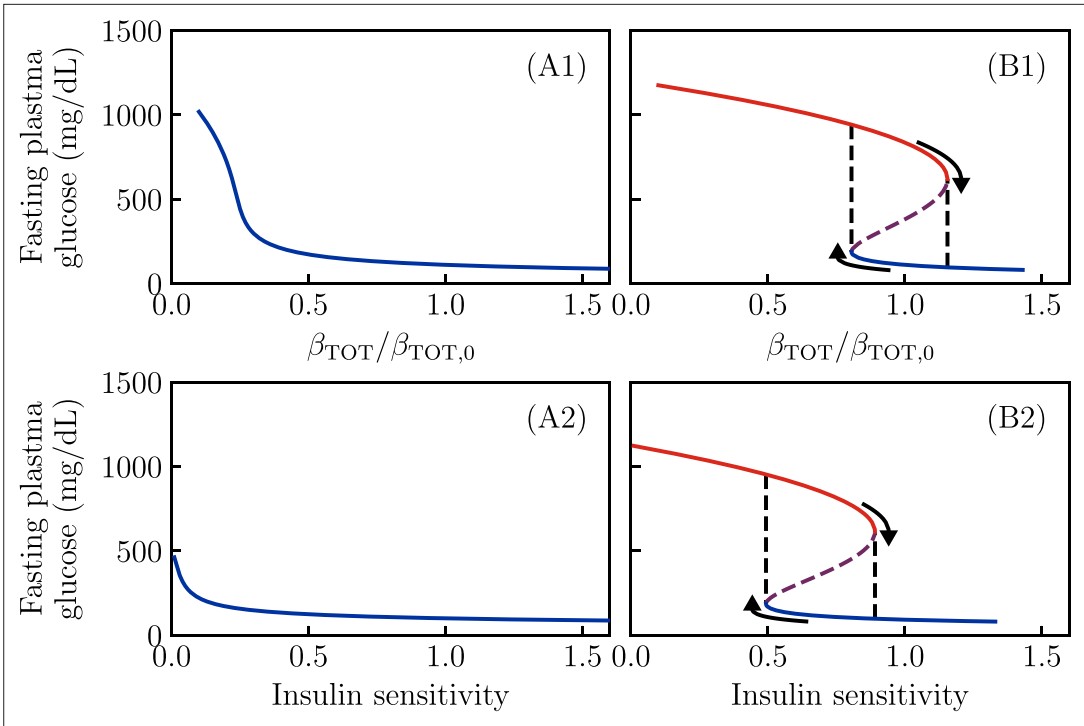

**Figure 5.** Fixed points as a function of total β-cell population and insulin sensitivity. All other parameters of the model are taken to be as in *Figure 2A and B*. (**A1, A2**) Regardless of insulin sensitivity or total β-cell number, only a single stable value of fasting glucose is possible if the reversible -cell deactivation rate is low. (**B1**) For a set of parameters which gives bistability, decreasing the total number of β-cells removes the high-$\beta$ fixed point, producing total insulin dependence. (**B2**) An improvement in insulin sensitivity can push the system past a bifurcation as well, removing the low-$\beta$ fixed point and thus producing partial remission.

We now go on to study the full plane of possible $(S_I, \beta_{\mathrm{TOT}})$ values. For each value of $(S_I, \beta_{\mathrm{TOT}})$, we classify the resulting state based on the fixed points of fasting glucose, analyzed as in *Figure 3*. The result of this classification is shown in *Figure 6*. The color indicates the lowest value of fasting glucose at a given pair $(S_I, \beta_{\mathrm{TOT}})$, with blue indicating normal blood glucose, red indicating glucose above the ADA fasting glucose criterion for diabetes, and a gradient for values in between. The hatching indicates the presence of β-cell deactivation, with single hatching indicating the bistability that allows for entry into a reversible state of ketosis, and double hatching indicating that the high-$\beta$ fixed point has disappeared and β-cell function cannot be restored by insulin treatment (unless treatment also improves insulin sensitivity).

We see in *Figure 6* that the boundaries of the hatched regions roughly follow the contours of fasting glucose. Thus, even when the rate of irreversible inactivation is high enough to predispose to KPD, if the total β-cell population or $S_I$ is high enough to result in normal fasting glucose, this protects against the $\beta$-inactive state. Note, however, that the boundary of the bistable region is at higher fasting glucose for larger $S_I$ (which increases the strength of feedback of β-cell deactivation on glucose).

At fixed $\beta_{\mathrm{TOT}}$, greater insulin resistance both predisposes to KPD in this model and will result in higher glucose levels after treatment, consistent with clinical data (*Table 1, criteria X Y*). However, note that the observed association of insulin sensitivity with remission is without controlling for β-cell function and thus may be confounded.

Finally, we note that, if the parameters of this model could be accurately measured in a patient, then the model would predict the course of remission for a given insulin treatment schedule, and thus assist in designing treatment. In this model, tighter insulin control will always produce a faster remission. Thus, the optimal treatment protocol (assuming glucose consumption is minimized) is to pick a safe target fasting glucose which avoids the risk of hypoglycemia, and then titrate the insulin dose to achieve this target glucose.

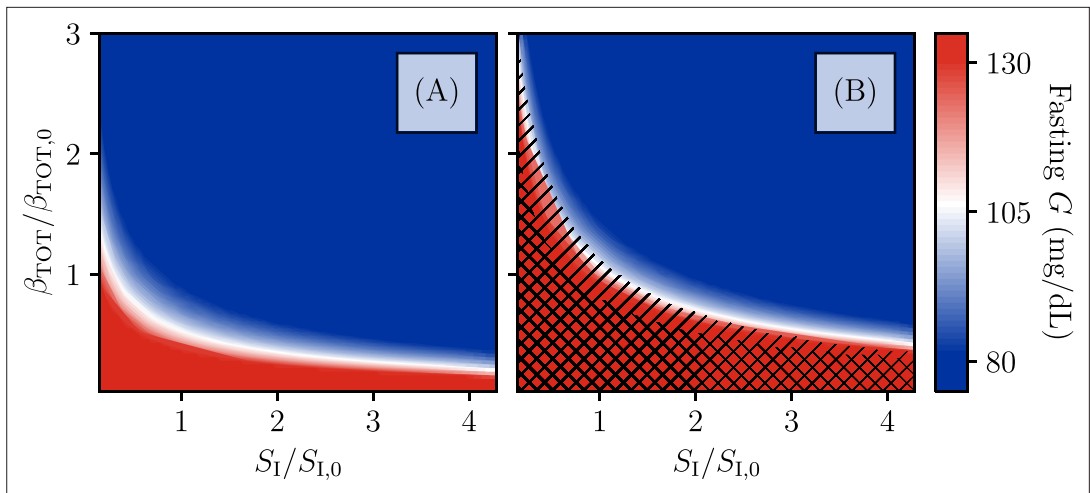

**Figure 6.** Phase diagram of the model. Fixed-point structure of the model as a function of total β-cell population and insulin sensitivity $S_1$, relative to the values used in *Figure 2A and B*. Color indicates fasting glucose (with all values of $G$ above 130 mg/dL the same red), while hatching indicates the number of fixed points. No hatching indicates existence of just one stable fixed point at high $\beta$ and intermediate or normal fasting glucose. Single-hatched regions have two fixed points, allowing for transient loss of β-cell function and remission, while double-hatched regions have only the low-β fixed point (i.e., total insulin dependence). (**A**) For the -cell inactivation rate as in *Figure 2A*, the bistability which produces ketosis-prone diabetes mellitus (KPD)–like presentation and remission is not seen, even for β-cell populations and $S_1$ values which produce fasting hyperglycemia consistent with T2D. (**B**) For the β-cell inactivation rate as in *Figure 2B*, mild fasting hyperglycemia tends to correspond with a susceptibility to KPD. Reduction of β-cell populations (e.g., due to the slower, permanent glucotoxicity as in *Figure 2D*) can produce greater hyperglycemia and even total insulin dependence. Improvements in insulin sensitivity, in contrast, contribute to the maintenance of remission.

Assuming no sugar consumption, the details of this protocol can easily be calculated for a given set of parameters in this model ('Materials and methods'). The result is shown in *Figure 7* for the parameters of the KPD patient in *Figure 2A*. Panel A shows how the time to achieve remission, here defined as the time to reach the stable fixed point value of $\beta$, depends on the targeted fasting glucose $G_{\min}$ during insulin treatment. Remission is only achieved if a value lower than the stable fixed-point value (which, for this patient, is 110 mg/dL) is targeted, and lower glucose targets produce a faster

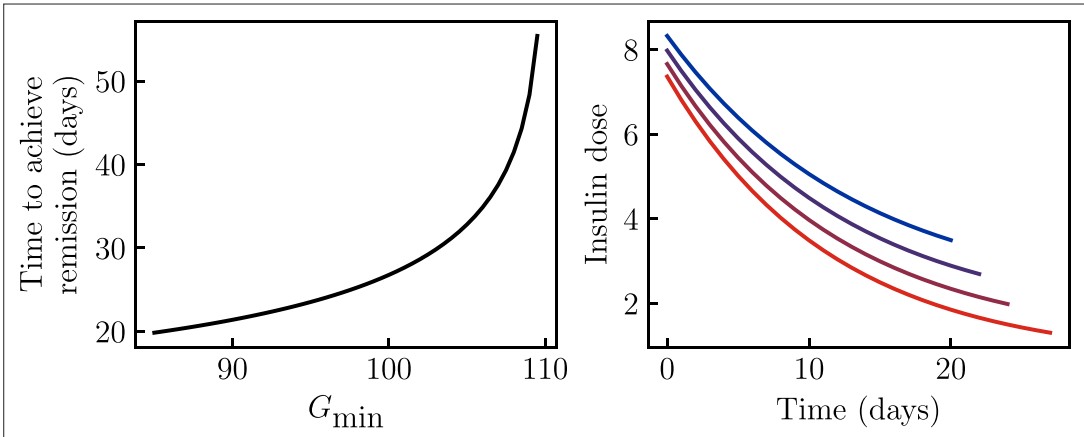

**Figure 7.** Optimal insulin treatment protocol. Parameters of the model are taken to be those in *Figure 2B*, and we determine the optimal treatment protocol under the assumptions that the patient does not consume sugar, and that fasting glucose must not be allowed below some target value $G_{\min}$ to avoid the risk of hypoglycemia. (**A**) The time to achieve remission becomes longer as the target becomes higher, and remission is never achieved if the target $G_{\min}$ is below the fixed point value. (**B**) The optimal insulin treatment protocol under these assumptions, for $G_{\min} = 85, 90, 95, 10$ (blue to red).

remission. Alternative definitions of the time to remission (e.g., crossing the unstable fixed point $\beta$ value) will produce qualitatively similar results. Panel B shows the insulin treatment protocol that maintains this target fasting glucose, for several such targets. We see that, because of the improvement in β-cell function over the course of treatment, the model predicts that achieving a given target requires an insulin dose that starts high and is decreased exponentially over time.

## Discussion

Our model predicts that the primary difference between individuals predisposed to KPD and those who are predisposed to 'conventional' T2D are the rates associated with the reversible glucotoxicity process (through their ratio, $k_{IN}/k_{RE}$). Other parameters, such as the total β-cell population and insulin resistance, are predicted to be associated with both KPD and 'conventional' T2D. In particular, since a lower total β-cell reserve is associated with both the appearance of KPD and difficulty in achieving remission, a period of uncontrolled T2D, even conventional T2D, during which the slow glucotoxicity will reduce the number of β-cells, can lead to the onset of KPD and is predicted to make recovery less likely.

From KPD patient to KPD patient, we expect all parameters of our model to vary. In *Figure 2*, we explicitly showed that variations in $k_{RE}$ can change the time needed to achieve remission, and that variations in $k_D$ can prevent remission from ever being achieved, but individual KPD patients are also expected to differ in $k_{IN}$, as well as in insulin sensitivity, glucose effectiveness, etc. In fact, since glucotoxicity is presumably the result of a complex pathway rather than a simple biochemical reaction, even the functional form of $g(G)$ may vary between individuals. Practical usage of this model in the clinic will require both assessing which of these parameters vary substantially in the population and developing protocols to measure those which vary in individual patients.

Although KPD is more prevalent in specific ethnic groups, very little is known about specific genetic factors which predispose individuals to KPD. A *pax4* allele specific to West African populations seems to be associated with KPD, with a small number of monozygous individuals found in a sample of KPD patients and none in control individuals or those with conventional T2D (*Mauvais-Jarvis et al., 2004a*). Since PAX4 is expressed during the development of pancreatic islets, where β-cells are located, it was hypothesized that this variant acts either through an effect on the total number of β-cells formed or through the formation of defective -cells (*Mauvais-Jarvis et al., 2004a*). Since, in our model, a decrease in the total number of β-cells should predispose to either KPD or 'ordinary' T2D equally well, depending on the rate of the reversible inactivation process $k_{IN}$, our results would suggest that this allele might predispose to KPD through a β-cell phenotype that is more susceptible to reversible inactivation. Other work has found that KPD is associated with deficiency in glucose-6-phosphate dehydrogenase, an enzyme which plays a role in reducing oxidative stress, which is consistent with a mechanism where this reversible glucotoxicity is mediated by oxidative stress (*Sobngwi et al., 2005*). The fact that KPD patients tend to have a family history of diabetes (*Umpierrez et al., 1995*) is also consistent with genetic variation in the rate of reversible inactivation of β-cells.

If, as we hypothesize, the reversible activation process is present in all individuals, with a heterogeneous rate, then it is possible that 'conventional' T2D always involves some degree of reversible inactivation, with KPD representing only the extreme end of a continuum. This may explain observations of partial remission after insulin treatment in populations with more conventional forms of T2D (*Ryan et al., 2004*; *Weng et al., 2008*; *Retnakaran et al., 2010*; *Retnakaran et al., 2014*; *Kramer et al., 2013a*; *Kramer et al., 2013b*; *Chon et al., 2018*). In fact, others have also argued that these data are evidence of the reversal of glucotoxicity (*Retnakaran et al., 2024*).

We might also expect that other situations involving transient improvements in diabetes symptoms may be analyzed using this model or a similar one. Although T1D is primarily driven by immune destruction of β-cells, the 'honeymoon phase', where symptoms temporarily improve upon insulin treatment, may involve some component of glucotoxicity reversal. Further, instances of diabetes remission produced by dramatic changes in insulin resistance (such as recovery from gestational diabetes postpartum, or reversal of T2D through very low-calorie diets) may also involve some component of glucotoxicity reversal. We plan to study whether our model can account in detail for these cases and discuss them in a future manuscript. Other phenomena, like the rapid (and difficult-to-reverse) progression of T2D in youth (*Utzschneider et al., 2021*), may possibly be understood using

models like the one discussed here through person-to-person variation in the rate of *permanent* β-cell deactivation.

We have kept this preliminary model simple in order to show which features are essential in order to produce the phenomena of KPD. Many details we have omitted, however, may play an important role in analyzing real patient data. Some of the omitted details, and the effects we expect them to have, are

1. Our description of glucose intake in meals has been simplistic. This does not affect the predictions of whether or not KPD-like symptoms are possible for given parameters, but means we may be underestimating or overestimating the amount of glucose that must be consumed to drive switching to the low-$\beta$ state.
2. Similarly, we have neglected the well-known time delay of insulin action present in the commonly-used minimal model of glucose kinetics (*Bergman et al., 1979*), which only affects fast timescale dynamics, and thus may adjust the effect predicted for a specific meal.
3. We lack a realistic description of kidney clearance of glucose above a threshold (*De Gaetano et al., 2014*). Including this would likely result in a lower predicted glucose in the low-$\beta$ state and could even prevent the appearance of the low-$\beta$ state for some parameter values. On the other hand, it will have little effect on the dynamics once treatment begins.
4. Our simple model of β-cell adaptation neglects the known hyperglycemia-induced leftward shift in the insulin secretion curve ($f(G)$ in *Equation 2*; *Ha et al., 2016*; *Glynn et al., 2016*; *Leahy et al., 1993*). In our model, bistability requires a region where (after adaptation and glucotoxicity equilibrate) higher glucose reduces insulin secretion. Thus, allowing for this adaptation of $f(G)$, by producing an earlier saturation of $f(G)$, would reduce the threshold of $k_{IN}/k_{RE}$ required to produce KPD-like symptoms. But this could not, by itself, produce these symptoms without any reversible glucotoxicity. Further, this effect will likely need to be included to quantitatively model remission since existing data already suggest changes in the shape of $f(G)$ over the course of remission (*Umpierrez et al., 1995*).

A period of extreme hyperglycemia and low insulin secretion does not guarantee the presence of ketoacidosis. Thus, our model is consistent with the observation that a similar remission is seen in insulin treatment of patients with similar demographics to KPD patients who present with severe hyperglycemia, but not ketoacidosis (*Umpierrez et al., 1995*).

How could this model of KPD be more fundamentally incorrect? Broadly, there are two possible classes of alternatives. Firstly, the pathology of KPD may not actually reflect bistability. Secondly, bistability could be generated by a different mechanism.

The first type of alternative would be one in which the decline in β-cell function preceding presentation is not actually that rapid, and instead some other change prompts the apparently rapid clinical presentation. For example, chronically high glucose could drive dehydration, worsening kidney function and thus leading to both further increases in glucose and inability to clear ketones.

What other mechanisms could generate bistability on a suitable timescale? There are some reversible processes which may be suitable for this role. The adaptive change in the shape of $f(G)$ is believed to occur over the scale of weeks (*Ha et al., 2016*), and in animal models, changes in diet can produce insulin resistance on timescales of weeks (*Mittelman et al., 2000*). It has been proposed in the past that insulin resistance may adaptively change in response to levels of insulin or glucose (*Rossetti et al., 1987*; *DeFronzo, 1988*; *Ader et al., 2014*). Furthermore, insulin resistance and the shape of $f(G)$ can both be seen to improve during the remission of KPD (*Umpierrez et al., 1995*). Thus, a feedback loop with a suitable timescale could exist between changes in insulin secretion and insulin resistance or glucose effectiveness. On the other hand, these changes are generally thought to be adaptive, and thus it is unclear how they would form a positive feedback loop rather than a negative feedback loop.

One final possibility is that insulin per se acts to produce remission. Although insulin treatment is associated with reductions in insulin resistance (*Umpierrez et al., 1995*; *Scarlett et al., 1982*), this effect is likely downstream of glucose levels: insulin itself has been proposed to promote insulin resistance (*Ader et al., 2014*; *DeFronzo, 1988*). Thus, if remission is produced by insulin per se rather than normoglycemia, it would likely be mediated by insulin acting directly on β-cells to promote insulin secretion, as suggested, for example, by experiments where β-cell specific insulin receptor knockouts show a reduction in glucose-stimulated insulin secretion (*Kulkarni et al., 1999*). On the other hand,

due to the much lower relative concentration of insulin produced in the islets by exogenous insulin administration rather than endogenous insulin (*Unger and Cherrington, 2012*), this seems unlikely.

The qualitative success of our model suggests that multiple distinct types of glucotoxicity may operate in the pathogenesis of T2D and KPD: a slower, irreversible death of β-cells which is responsible for slow, irreversible decline in T2D and failure to maintain or achieve remission in KPD, and a faster, reversible process associated with rapid onset and remission of KPD. The clearest test of our model would be a cellular or systems-biology-based identification and characterization of multiple β-cell states and inactivation mechanisms, all dependent on glucose level. Some pioneering studies along these lines have been done already, showing diversity of β-cell secretory phenotypes and transition between phenotypes in a glucose-dependent way (*Miranda et al., 2021*; *Laybutt et al., 2002*; *Jonas et al., 1999*; *Laybutt et al., 2003*; *Wang et al., 2014*; *Dahan et al., 2017*), though mostly in rodent models. Such experiments show that roughly 2–4 weeks of hyperglycemia can change gene expression and secretory phenotypes in β-cells, and then normoglycemia can restore normal function on similar scales (*Laybutt et al., 2002*; *Jonas et al., 1999*; *Laybutt et al., 2003*). Thus, there is evidence that reversible glucotoxicity, of a type similar to what we have described, exists in rodents. However, further work should attempt to establish the glucose-dependent rate of this loss of function more quantitatively. Further, lab-based cellular-level characterization of similar phenomena in human β-cells and in pancreatic islets, or analysis of human physiological data still must be done. Crucially, it remains to be seen if the rates of β-cell activation and deactivation in such experiments in humans would correlate with the predisposition to KPD, which is a clear prediction of our model.

Our qualitative results do not require that $g(G)$ has the precise form that we have assumed, but we can place some constraints on its form. Producing bistability requires that, after allowing glucotoxicity processes to equilibrate, total insulin secretion *decreases* as glucose is increased, for some values of glucose. This strongly suggests that $g(G)$ must rise substantially at large values of $G$, where $f(G)$ is plateauing, for our model to be correct. This prediction can be tested in more quantitative in vitro experiments.

As mentioned previously, *Umpierrez et al., 2007* found that a 20-hour infusion of glucose produced a roughly 50% decline of β-cell function in a KPD patient. During this infusion, $G$ fluctuated around 150 mg/dL. The functional form of $g(G)$ and rate constant $k_{IN}$ used as the rate of the reversible reactivation rate gives a roughly 10% decline of $\beta$ per day at 150 mg/dL; given the nonlinearity of $g(G)$ this is an underestimate of the decline produced by $G$ which fluctuates substantially around 150 mg/dL as in the experiment. Thus, our rate of reversible inactivation is roughly consistent with the limited existing data in KPD patients. However, given that this data comes from a single patient and that we expect the rate to vary substantially from patient-to-patient, many more such in vivo measurements are needed to understand the reversible deactivation process in a whole body. If $G$ can be well controlled in a clamp, such measurements could also shed light on the functional form of $g(G)$.

This general class of models, even if the functional form of $g(G)$ that we have assumed is not accurate, makes several predictions about the natural history of KPD which can be tested. As seen in *Figure 6*, our model generally produces bistability only when glucose is already at least slightly elevated. In other words, within our model, the onset of KPD crucially depends on the β-cell adaptation mechanisms having reached nearly maximum adaptation, and then the patient undergoing a period of high carbohydrate consumption. While this is anecdotally true in our practice (*Table 1, criterion 3*), there is no strong quantitative data yet to support this requirement of the model. Thus, possibly the easiest way of validating or disproving our model is through focusing on careful collection of quantitative antecedent data at presentation on the recent food intake history and HbA1c (as a proxy for hyperglycemia over the preceding few months), as well as HbA1c at previous routine screenings, where available.

One clear prediction of the model, at least with the form of $g(G)$ we have considered, is that recovery of glucose control is gradual. This should be testable using more frequent measurements of glucose, as well as tracking of the needed insulin doses, during the process of remission.

Further, in order to constrain the functional form of $g(G)$ in models of this type, more clinical data will be necessary. Most past work has not analyzed remission with this frequency, instead tracking clinical progress every few weeks or months (*Mauvais-Jarvis et al., 2004b*; *Umpierrez et al., 1995*; *Vellanki et al., 2016*). The form of $g(G)$, however, cannot be well constrained by low-frequency measurements such as these. As seen in *Figure 2B and C*, in this model, patients with different values of the reversible

inactivation rate $k_{IN}$ and reactivation rate $k_{RE}$ can still present with similar levels of hyperglycemia and achieve similar final states upon remission. This suggests that new, higher frequency data should be collected in order to better distinguish possible mechanisms of KPD pathogenesis using models of this type. In particular, continuous glucose monitoring would allow the dynamics of remission to be studied, which would provide a much more stringent test of the form of $g(G)$.

These data will also constitute a test of the model by testing quantitative predictions about the improvement of β-cell function over time during treatment. The best test of the model, however, would come from comparing such data to measurements of $g(G)$ in vitro, of the type discussed above.

Current long-term treatment for KPD is extrapolated from T2D standards of care. Even without quantitative fits to specific patients, our model suggests KPD-specific management strategies. Thus, it can inform clinical trials that may lead to specific clinical recommendations for KPD. For example: lifestyle guidance derived from T2D standards of care focuses on weight management. In our model, since $S_I$ effects KPD just as strongly as conventional T2D, we expect that such guidance is, indeed, valuable. Our model, however, would suggest that limiting carbohydrates per se, as in T1D, would also be advisable. Further, due to the importance of sustained high blood glucose to cross the $\beta$ threshold and switch fixed points, perhaps KPD patients should be cautioned that if they consume a lot of sugar one day, it is safest to lower their sugar intake for the subsequent few days.

In the longer term, patient-specific fits of a more detailed version of this model could provide personalized clinical guidance on insulin treatment, as suggested by *Figure 7*. Such guidance could allow for remission to be achieved quickly without giving too high an insulin dose (risking hypoglycemia) or excessive (costly) doctor–patient interactions.

While simple models of the type that we developed here are certainly wrong in that they do not account for the full complexity of the underlying physiology, they can be useful both for basic biomedical science and in the clinic. Specifically, they predict new β-cell states and activation/deactivation processes, and hence tell us to look for diversity of β-cell phenotypes. They also suggest that clinical practice should be modified to include more frequent follow-ups (or continuous monitoring) to generate data on temporal scales that match the dynamics of the model. And they propose key mechanisms (rates of secretory phenotype switching) that might determine predisposition to KPD, which also can be accessed experimentally or even clinically. Finally, the simplicity of such models makes it plausible that the proposed additional feedback loops in the glucose control circuitry may be general enough to contribute to the explanation of other observations that involve rapid onset or remission of diabetes-like phenomena, as discussed above.

## Materials and methods
### Model
We model the short timescale of dynamics of glucose and insulin as

$$\frac{dG}{dt} = M(t) + \frac{m_0}{I_0 + I} - (S_E + S_I I)G, \tag{1}$$

$$\frac{dI}{dt} = \beta c f(G) - \gamma I, \tag{2}$$

$M(t)$ represents the intake of glucose in meals, while $\frac{m_0}{I_0 + I}$ represents the endogenous production of glucose, mainly in the liver. The final terms represent both insulin-independent and insulin-dependent clearance of glucose. *Ha and Sherman, 2020* uses a similar model of HGP suppression by insulin, while this glucose production term is instead a constant $R_0$ in *Topp et al., 2000*, *Karin et al., 2016*, and *Ha et al., 2016*.

Here $f(G)$ is the β-cell insulin secretion rate (normalized by the blood volume), which we take to be $f(G) = G^2/(G^2 + K_f^2)$, as in *Topp et al., 2000* and *Karin et al., 2016*. $c$ is a measure of the mass or secretory activity of an individual β-cell, which we assumed to adapt to produce a target fasting glucose as described below. Insulin is assumed to be cleared from the blood by the liver at a constant rate $\gamma$.

We model β-cell dynamics using

$$\frac{d\beta}{dt} = -\big(k_{IN}g(G) + k_D h(G)\big)\beta + k_{RE}\beta_{IN} \tag{3}$$

$$\frac{\mathrm{d}\beta_{\mathrm{IN}}}{\mathrm{d}t} = -\left(k_{\mathrm{RE}} + k_{\mathrm{D}}h(G)\right)\beta_{\mathrm{IN}} + k_{\mathrm{IN}}g(G)\beta. \tag{4}$$

Here $k_{\mathrm{D}}h(G)$ is the β-cell death rate, $k_{\mathrm{RE}}$ is the (glucose-independent) β-cell reactivation rate, and $k_{\mathrm{IN}}g(G)$ is the rate of reversible β-cell inactivation. In all simulations, we take $h(G) = G^2/(G^2 + K_h^2)$ and $g(G) = G^2/(G^2 + K_g^2)$ with $K_g$ chosen to be very large since generating bistability requires a substantial drop in steady-state insulin secretion at high steady-state glucose, and thus requires $K_g \gg K_f$.

We also assume that β-cell mass, through $c$, adapts to produce a target fasting glucose $G_0$ (*Topp et al., 2000*; *Karin et al., 2016*; *Ha et al., 2016*), which we take to be 80 mg/dL. We assume that this adaptation has a maximum capacity (i.e., a maximum value of $c$) (*Woller et al., 2024*). Thus,

$$\frac{\mathrm{d}c}{\mathrm{d}t} = a(G_f - G_0, c), \tag{5}$$

for some function $a$ satisfying:

1. $a > 0$ when $G_f > G_0$, $a < 0$ when $G_f < G_0$, and $a = 0$ when $G_f = G_0$, for $c < c_{\max}$.
2. $a = 0$ when $c = c_{\max}$.

Thus, steady-state requires that either $G_f = G_0$, or $G_f > G_0$ and $c = c_{\max}$.

In *Figures 2–5*, we have assumed chronic hyperglycemia, and thus $c = c_{\max}$ does not change. In the analysis of *Figure 6*, $c$ remains at its maximum value whenever $G > 80$ mg/dL. If, however, $S_I$ or $\beta_{\mathrm{TOT}}$ is large enough to allow $G < 80$ mg/dL, then $c$ decreases to the steady state $G_f = 80$ mg/dL.

## Simulations

The parameters $\gamma$, $S_E$, and $S_I$ are taken from *Topp et al., 2000*. *Topp et al., 2000*, *Karin et al., 2016*, and *Ha et al., 2016* take the rate of hepatic glucose production (HGP) to be a constant while we allow it to decrease when insulin increases, similarly to *Ha and Sherman, 2020*. We chose $I_0 = 5\,\mu\mathrm{U\,mL}^{-1}$, which is on the scale of observed fasting insulin values, and chose $m_0$ so that our rate of HGP matches the constant $R_0$ used in *Topp et al., 2000*, *Karin et al., 2016*, and *Ha et al., 2016* when $I = I_0$. For the chosen level of hyperglycemia and other parameters, this produces a fasting insulin of roughly $7\,\mu\mathrm{U\,mL}^{-1}$ and a fasting HGP roughly 15% lower than the constant $R_0$ used by *Topp et al., 2000*, *Karin et al., 2016*, and *Ha et al., 2016*.

During 'normal' periods, the sugar consumption from the three 'meals' adds up to roughly 3% of the daily HGP, while during the high sugar consumption period it adds up to roughly 70% of the daily HGP. In humans, 2 mg/(kg min) is a typical glucose flux (*DeFronzo, 1988*). Assuming a body weight of 70 kg, this amounts to 201 g of glucose per day, so that 70% of this would correspond to 3–4 cans of soda per day, a plausible 'high sugar consumption' condition. However, note that the rate of glucose appearance from *Topp et al., 2000*, interpreted in recent works as the rate of glucose production (*Ha and Sherman, 2020*; *Karin et al., 2016*), is actually about 1/7 of this estimate (assuming a plasma volume of 3.5 L). Indeed, *Topp et al., 2000* originally interpreted this term as the difference of the rate of glucose production and a glucose-independent component of the rate of glucose disposal. Thus, while the scale of sugar consumption relative to HGP required to produce the onset of DKA in our model is reasonable, the parameters will likely need some adjustment when fitting to patient data.

The rates of the hypothesized reversible glucotoxicity processes were then chosen by trial and error to reproduce the desired variation of possible patient phenotypes.

All numerical simulations are done using the differential equations package Diffrax (*Kidger, 2021*). We used a fifth-order implicit Runge–Kutta method due to *Kværnø, 2004*, adjusting the timestep with a PID controller (*Hairer and Wanner, 1991*; *Söderlind, 2002*).

## Dynamical systems analysis

To analyze the possible stable equilibria at a given, fixed exogenous glucose intake rate $M$ and exogenous insulin flux $F_I$, we assume equilibration of the fast dynamical equations, *Equations 1 and 2*. The steady-state glucose $G$ then solves the equation

$$\frac{m_0}{(\beta cf(G) + F_I)/\gamma + I_0} + M = G\left(S_E + \frac{S_I(\beta cf(G) + F_I)}{\gamma}\right). \tag{6}$$

Multiplying by $(\beta c f(G) + F_I)/\gamma + I_0$, and then subsequently by the denominator of the Hill function $f(G)$, yields a polynomial equation for $G$, which we solve using the companion matrix (**Horn and Johnson, 1985**; **Harris et al., 2020**).

The steady-state insulin is then $(\beta c f(G) + F_I)/\gamma$, and the steady-state $\beta$ is $\beta_{TOT}/(1 + (k_{IN}/k_{RE})g(G))$.

## Optimal insulin protocol

In our model, remission is faster at lower glucose, so that, naively, one would choose a clinical intervention that would lower glucose the fastest.

However, this does not account for the possibility of hypoglycemia. We do not model it directly, but, in practice, fear of hypoglycemic episodes (due to greater volatility of blood glucose in reality than in our model) limits the aggressiveness of insulin treatment. Thus, we assume there is some target $G_{min}$ that the physician does not wish to lower the fasting glucose below.

Under this assumption, the fastest remission is produced by zero sugar consumption and an insulin dose which is chosen to always keep $G = G_{min}$. In reality, long-acting insulin is taken, for example, once per day, but for a simple analysis of the dose schedule over weeks we neglect this and assume the insulin flux can be continuously adjusted.

Under this assumption, the dynamics can be solved exactly by substituting $G(t) = G_{min}$ in **Equations 3 and 4** (again neglecting permanent glucotoxicity). The result is

$$\frac{\beta(t)}{\beta_{TOT}} = e^{-(k_{RE}+k_{IN}g(G_{min}))t} \left( \frac{\beta(0)}{\beta_{TOT}} - \frac{k_{RE}}{k_{RE} + k_{IN}g(G_{min})} \right) + \frac{k_{RE}}{k_{RE} + k_{IN}g(G_{min})}. \tag{7}$$

**Equation 1** then reduces to a quadratic equation for the necessary $I(t)$ to maintain $G(t) = G_{min}$, and then **Equation 2** can be solved for the necessary insulin dosage $F_I(t)$ to maintain this $I(t)$.

## Acknowledgements

We thank Larry S Phillips, Venkat Narayan, Guillermo E Umpierrez, and Darko Stefanovski for useful discussions. IN and SAR were supported, in part, by the Simons Foundation Investigator Program grant to IN, and SAR was additionally supported by Emory Global Diabetes Research Center and Emory University Research Council grants to IN and PV. PV was supported, in part, by the NIH through grants 1K23DK113241 and 1R03DK129627. We acknowledge support of our work through the use of the HyPER C3 cluster of Emory University's AI.Humanity Initiative.

---

## Additional information

### Competing interests

Priyathama Vellanki: Consultant for Eli Lilly, July 2023. The other authors declare that no competing interests exist.

### Funding

| Funder | Grant reference number | Author |
|---|---|---|
| Simons Foundation | MPS-SIP-00827661 | Ilya Nemenman |
| Emory Global Diabetes Research Center | | Priyathama Vellanki Ilya Nemenman |
| Emory University Research Council | | Priyathama Vellanki Ilya Nemenman |
| National Institute of Diabetes and Digestive and Kidney Diseases | 1R03DK129627-01A1 | Priyathama Vellanki |
| National Institute of Diabetes and Digestive and Kidney Diseases | 1K23DK113241-01A1 | Priyathama Vellanki |

| Funder | Grant reference number | Author |
|--------|------------------------|--------|

The funders had no role in study design, data collection and interpretation, or the decision to submit the work for publication.

## Author contributions

Sean A Ridout, Conceptualization, Formal analysis, Investigation, Methodology, Writing – original draft, Writing – review and editing; Priyathama Vellanki, Ilya Nemenman, Conceptualization, Supervision, Funding acquisition, Methodology, Writing – original draft, Writing – review and editing

## Author ORCIDs

Sean A Ridout ⓘ https://orcid.org/0000-0003-2387-8361
Priyathama Vellanki ⓘ https://orcid.org/0000-0002-6544-015X
Ilya Nemenman ⓘ https://orcid.org/0000-0003-3024-4244

Reviewer #1 (Public review): https://doi.org/10.7554/eLife.100193.3.sa1
Reviewer #2 (Public review): https://doi.org/10.7554/eLife.100193.3.sa2
Author response https://doi.org/10.7554/eLife.100193.3.sa3

## Additional files

### Supplementary files
MDAR checklist

### Data availability
All simulation and analysis code used to produce all data in this article, as well as code used to produce all figures in this article, is available at Zenodo.

The following dataset was generated:

| Author(s) | Year | Dataset title | Dataset URL | Database and Identifier |
|-----------|------|---------------|-------------|-------------------------|
| Ridout S | 2025 | Saridout/kpd_model_code: Revisions of manuscript | https://zenodo.org/records/15272218 | Zenodo, 10.5281/zenodo.15272218 |

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
