## [Editor Report · eLife Assessment]

This theoretical study makes a **useful** contribution to our understanding of a subtype of type 2 diabetes – ketosis-prone diabetes mellitus (KPD) – with a potential impact on our broader understanding of diabetes and glucose regulation. The article presents an ordinary differential equation-based model for KPD that incorporates a number of distinct timescales – fast, slow, as well as intermediate, incorporating a key hypothesis of reversible beta cell deactivation. The presented evidence is **solid** and shows that observed clinical disease trajectories may be explained by a simple mathematical model in a particular parameter regime.

---

## [Referee Report · Reviewer #1 (Public review)]

The goal of this work is to understand the clinical observation of a subgroup of diabetics who experience extremely high levels of blood glucose levels after a period of high carbohydrate intake. These symptoms are similar to the onset of Type 1 diabetes but, crucially, have been observed to be fully reversible in some cases.

The authors interpret these observations by analyzing a simple yet insightful mathematical model in which β-cells temporarily stop producing insulin when exposed to high levels of glucose. For a specific model realization of such dynamics (and for specific parameter values) they show that such dynamics lead to two distinct stable states. One is the relatively normal/healthy state in which β-cells respond appropriately to glucose by releasing insulin. In contrast, when enough β-cells "refuse" to produce insulin in a high-glucose environment, there is not enough insulin to reduce glucose levels, and the high-glucose state remains locked in because the high-glucose levels keep β-cells in their inactive state. The presented mathematical analysis shows that in their model the high-glucose state can be entered through an episode of high glucose levels and that subsequently the low-glucose state can be re-entered through prolonged insulin intake.

The strength of this work is twofold. First, the intellectual sharpness of translating clinical observations of ketosis-prone type 2 diabetes (KPD) into the need for β-cell responses on intermediate timescales. Second, the analysis of a specific model clearly establishes that the clinical observations can be reproduced with a model in which β-cells dynamics reversibly enter a non-insulin-producing state in a glucose-dependent fashion.

The likely impact of this work is a shift in attention in the field from a focus on the short and long-term dynamics in glucose regulation and diabetes progression to the intermediate timescales of β-cell dynamics. I expect this to lead to much interest in probing the assumptions behind the model to establish what exactly the process is by which patients enter a 'KPD state'. Furthermore, I expect this work to trigger much research on how KPD relates to "regular" type 2 diabetes and to lead to experimental efforts to find/characterize previously overlooked β-cell phenotypes.

In summary, the authors claim that observed clinical dynamics and possible remission of KPD can be explained through introducing a temporarily inactive β-cell state into a "standard model" of diabetes. The evidence for this claim comes from analyzing a mathematical model and clearly presented.

---

## [Referee Report · Reviewer #2 (Public review)]

In this manuscript, Ridout et al. present an intriguing extension of beta cell mass-focused models for diabetes. Their model incorporates reversible glucose-dependent inactivation of beta cell mass, which can trigger sudden-onset hyperglycemia due to bistability in beta cell mass dynamics. Notably, this hyperglycemia can be reversed with insulin treatment. The model is simple, elegant, and thought-provoking.

---

## [Author Response]

The following is the authors’ response to the original reviews.

**Public reviews:**
Concerning the grounding in experimental phenomenology, it would be beneficial to identify specific experiments to strengthen the model. In particular, what evidence supports reversible beta cell inactivation? This could potentially be tested in mice, for instance, by using an inducible beta cell reporter, treating the animals with high glucose levels, and then measuring the phenotype of the marked cells. Such experiments, if they exist, would make the motivation for the model more compelling.

There is some direct evidence of reversible beta cell inactivation in rodent / in vitro models. We had already mentioned this in the discussion, but we have added some text emphasizing / clarifying the role of this evidence (lines 359–362).

Others have also argued that some analyses of insulin treatment in conventional T2D, which has a stronger effect in patients with higher glucose before treatment, provides indirect evidence of reversal of glucotoxicity. We have also mentioned this in the revised paper (lines 284–285).

For quantitative experiments, the authors should be more specific about the features of beta cell dysfunction in KPD. Does the dysfunction manifest in fasting glucose, glycemic responses, or both? Is there a ”pre-KPD” condition? What is known about the disease’s timescale?

The answers to some of these questions are not entirely clear—patients present with very high glucose, and thus must be treated immediately. Due to a lack of antecedent data it is not entirely clear what the pre-KPD condition is, but there is some evidence that KPD is at least not preceded by diabetes symptoms. This point is already noted in the introduction of the paper and Table 1. However, we have added a small note clarifying that this does not rule out mild hyperglycemia, as in prediabetes (and indeed, as our model might predict) (lines 76–77). Similarly, due to the necessity of immediate insulin treatment, it is not clear from existing data whether the disorder manifests more strongly in fasting glucose or glucose response, although it is likely in both. (We might infer this since continuous insulin treatment does not produce fasting hypoglycemia, and the complete lack of insulin response to glucose shortly after presentation should produce a strong effect in glycemic response.) We believe our existing description of KPD lists all of the relevant timescales, however we have also slightly clarified this description in response to the first referee’s comments (lines 66–73, 83)

The authors should also consider whether their model could apply to other conditions besides KPD. For example, the phenomenology seems similar to the ”honeymoon” phase of T1D. Making a strong case for the model in this scenario would be fascinating.

This is an excellent idea, which had not occurred to us. We have briefly discussed this possibility in the remission (lines 281–291), but plan to analyze it in more detail in a future manuscript.

**Reviewer #1 (Recommendations for the author):**
Whenever simulation results are presented, parameter values should be specified right there in the figure captions.

We have added the values of glucotoxicity parameters to the caption of Figure 2. In other figures, we have explicitly mentioned which panel of Figure 2 the parameters are taken from. Description of the non-glucotoxicity parameters is a bit cumbersome (there are a lot of them, but our model of fast dynamics is slightly different from Topp et al. so it does not suffice to simply say we took their parameters) so we have referred the reader to the Materials and Methods for those.

I was confused by the language in Figure 4. Could the authors clarify whether they argue that: (1) the observed KPD behaviour is the result of the system switching from one stable state to another when perturbed with high glucose intake? (2) the observed KPD behaviour is the result of one of the steady states disappearing with high glucose intake?

What we mean to say is that during a period of high sugar intake or exogeneous insulin treatment, one of the fixed points is temporarily removed—it is still a fixed point of the “normal” dynamics, but not a fixed point of the dynamics with the external condition added. Since when glucose (insulin) intake is high enough, only the low (high)-β fixed point is present, under one of these conditions the dynamics flow toward that fixed point. When the external influx of glucose/insulin is turned off, both fixed points are present again—but if the dynamics have moved sufficiently far during the external forcing, the fixed point they end up in will have switched from one fixed point to the other. We have edited the text to make this clearer (lines 153–185). Do note, however, that in response to both referee’s comments (see below), Figures 3 and 4 have been replaced with more illuminating ones. This specific point is now addressed by the new Figure 3.

The adaptation of the prefactor ’c’ was confusing to me. I think I understood it in the end, but it sounded like, ”here’s a complication, but we don’t explain it because it doesn’t really matter”. I think the authors can explain this better (or potentially leave out the complication with ’c’ altogether?).

Indeed, the existence of an adaptation mechanism is important for our overall picture of diabetes pathogenesis, but not for many of our analyses, which assume prediabetes. Nonetheless, we agree that the current explanation of it’s role is confusing because of its vagueness. We have elaborated the explanation of the type of dynamics we assume for c, adding an equation for its dynamics to the “Model” section of the Materials and methods, explained in lines 456–465. We have also amended Figure 1 to note this compensation.

I expect the main impact of this work will be to get clinical practitioners and biomedical researchers interested in the intermediate timescale dynamics of β-cells and take seriously the possibility that reversible inactive states might exist. But this impact will only be achieved when the results are clearly and easily understandable by an audience that is not familiar with mathematical modelling. I personally found it difficult to understand what I was supposed to see in the figures at first glance. Yes, the subtle points are indeed explained in the figure captions, but it might be advantageous to make the points visually so clear that a caption is barely needed. For example, when claiming that a change in parameters leads to bistability, why not plot the steady state values as a function of that parameter instead of showing curves from which one has to infer a steady state?I would advise the authors to reconsider their visual presentation by, e.g., presenting the figures to clinical practitioners or biomedical researchers with just a caption title to test whether such an audience can decipher the point of the figure! This is of course merely a personal suggestion that the authors may decide to ignore. I am making this suggestion only because I believe in the quality of this work and that improving the clarity of the figures and the ease with which one can understand the main points would potentially lead to a much larger impact on the presented results.

This is a very good point. We have made several changes. Firstly, we have added smaller panels showing the dynamics of β to Figure 2; previously, the reader had to infer what was happening to β from G(t). Secondly, we have completely replaced the two figures showing dβ/dt, and requiring the reader to infer the fixed points of β, with bifurcation diagrams that simply show the fixed points of G and β. The new figures show through bifurcation diagrams how there are multiple fixed points in KPD, how glucose or insulin treatment force the switching of fixed points, and how the presence of bistability depends on the rate of glucotoxicity. (These new figures are Fig. 3–5 in the revised manuscript.)

Could the authors explicitly point out what could be learned from their work for the clinic? At the moment treatment consists of giving insulin to patients. If I understand correctly, nothing about the current treatment would change if the model is correct. Is there maybe something more subtle that could be relevant to devising an optimal treatment for KPD patients?

This is another very good point. We have added a new figure (Fig. 7) in our results section showing how this model, or one like it, can be analyzed to suggest an insulin treatment schedule (once parameters for an individual patient can be measured), and added some discussion of this point (lines 224–240) as well as lifestyle changes our model might suggest for KPD patients to the discussion (lines 413–425).

Similarly, could the authors explicitly point out how their model could be experimentally tested? For example, are the functions f(G) and g(G) experimentally accessible? Related to that, presumably the shape of those functions matters to reproduce the observed behaviour. Could the authors comment on that / analyze how reproducing the observed behaviour puts constraints on the shape of the used functions and chosen parameter values?

g(G) has not been carefully measured in cellular data, however it could be in more quantative versions of existing experiments. Further, our model indeed requires some general features for the forms of f(G) and g(G) to produce KPD-like phenomena. We have added some comment on this to the discussion section of the revised manuscript (lines 367–372).

Could the authors explicitly spell out which parameters they think differ between individual KPD patients, and which parameters differ between KPD patients and ’regular’ type 2 diabetics?

In general we expect all parameters should vary both among KPD patients and between KPD / “conventional” T2D. The primary parameter determining whether KPD and conventional T2D, is seen, however, is the ratio kIN/kRE. We have elaborated on both these points in the revised mansuscript. (Lines 186–192, 250–257.)

I was confused about the timescale of remission. At one point the authors write “KPD patients can often achieve partial remission: after a few weeks or months of treatment with insulin” but later the authors state that “the duration of the remission varies from 6 months to 10 years”.

The former timescale is the typical timescale achieve remission. After remission is reached, however, it may or may not last—patients may experience a relapse, where their condition worsens and they again require insulin. We have edited the text to clarify this distinction (lines 66–73).

When the authors talk about intermediate timescales in the main text could they specify an actual unit of time, such as days, weeks, or months as it would relate to the rate constants in their model for those transitions?

We have done so (lines 86–87, figure 1 caption, figure 2 caption). Getting KPD-like behavior requires (at high glucose) the deactivation process to be somewhat faster than the reactivation process, so the relevant scales are between weeks (reactivation) and days (deactivation at high G).

The authors state “Our simple model of β-cell adaptation also neglects the known hyperglycemiainduced leftward shift in the insulin secretion curve f(G) in Eq. 2 ”. This seems an important consideration. Could the authors comment on why they did not model this shift, and or explicitly discuss how including it is expected to change the model dynamics?

We agree that this process seems potentially relevant, as it seems to happen on a relatively fast timescale compared to glucose-induced β-cell death. It is, however, not so well characterized quantitatively that including it is a simple matter of putting in known values—we would be making assumptions that would complicate the interpretation of our results.

It is clear that this effect will need to be considered when quanitatively modelling real patient data. However, it is also straightforward to argue that this effect by itself cannot produce KPD-like symptoms, and will only tend to reduce the rate of glucotoxocity necessary to produce bibstability. We have added a discussion of this in the revisions (lines 307–315). We have also, in general, expanded the discussion of the effects that each neglected detail we have mentioned is expected to have (lines 292–315).

The authors end with a statement that their results may “contribute to explanation of other observations that involve rapid onset or remission of diabetes-like phenomena, such as during pregnancy or for patients on very low calorie diets.” Could the authors spell out exactly how their model potentially relates to these phenomena?

Our thinking is that, even when another direct cause, such as loss of insulin resistance, is implicated in reversal of diabetes, some portion of the effect may be explained by reversal of glucotoxicity. This is indeed at this point just a hypothesis, but we have expanded on it briefly in the revision. (Lines 281–291.)

Minor typos:In Figure 2.D the last zero of 200 on the axis was cut off.Line 359 - there is a missing word ”in the analysis”.

We have fixed these typos, thanks.

**Reviewer #2 (Recommendations for the author):**
The manuscript could be significantly improved in two key areas: the presentation of the analysis, and the relation with experimental phenomenology.Regarding the analysis presentation, the figures could be substantially enhanced with minimal effort from the authors. At present, they are sparse, lack legends, and offer only basic analysis. The authors should consider presenting, for example, a bifurcation diagram for beta cell mass and fasting glucose levels as a function of kIN, and how insulin sensitivity and average meal intake modulate this relationship. The goal should be to present clear, testable predictions in an intuitive manner. Currently, the specific testable predictions of the model are unclear.

The response to this question is copied from the reponses to related questions from the first referee.

This is a very good point. We have made several changes. Firstly, we have added smaller panels showing the dynamics of β to Figure 2; previously, the reader thad to infer what was happening to β from G(t). Secondly, we have completely replaced the two figures showing dβ/dt, and requiring the reader to infer the fixed points of β, with bifurcation diagrams that simply show the fixed points of G and β. The new figures show through bifurcation diagrams how there are multiple fixed points in KPD, how glucose or insulin treatment force the switching of fixed points, and how the presence of bistability depends on the rate of glucotoxicity. We have also supplemented our phase diagram that shows the effects of SI and the total beta cell population with bifurcation diagrams showing β as SI and βTOT are varied. (These new figures are Fig. 3–5 in the present manuscript.) Finally, we have added another figure analyzing the model’s predictions for the optimal insulin treatment and the resulting time needed to achieve remission (Fig. 7)